# Mice Lacking *Gpr179* with Complete Congenital Stationary Night Blindness Are a Good Model for Myopia

**DOI:** 10.3390/ijms24010219

**Published:** 2022-12-22

**Authors:** Baptiste Wilmet, Jacques Callebert, Robert Duvoisin, Ruben Goulet, Christophe Tourain, Christelle Michiels, Helen Frederiksen, Frank Schaeffel, Olivier Marre, José Alain Sahel, Isabelle Audo, Serge Picaud, Christina Zeitz

**Affiliations:** 1Sorbonne Université, INSERM, CNRS, Institut de la Vision, 75012 Paris, France; 2Service of Biochemistry and Molecular Biology, INSERM U942, Hospital Lariboisière, AP-HP, 75010 Paris, France; 3Department of Chemical Physiology & Biochemistry, Oregon Health & Science University, Portland, OR 97239, USA; 4Wavefront-Engineering Microscopy Group, Neurophotonics Laboratory, CNRS UMR8250, Paris Descartes University, 75270 Paris, France; 5Institute of Molecular and Clinical Ophthalmology Basel (IOB), 4056 Basel, Switzerland; 6Section of Neurobiology of the Eye, Ophthalmic Research Institute, University of Tuebingen, 72076 Tuebingen, Germany; 7Zeiss Vision Lab, Ophthalmic Research Institute, University of Tuebingen, 72076 Tuebingen, Germany; 8Department of Ophthalmology, The University of Pittsburgh School of Medicine, Pittsburgh, PA 15213, USA; 9Centre Hospitalier National d’Ophtalmologie des Quinze-Vingts, INSERM-DGOS CIC 1423, 75012 Paris, France; 10Fondation Ophtalmologique Adolphe de Rothschild, 75019 Paris, France; 11Académie des Sciences, Institut de France, 75006 Paris, France

**Keywords:** CSNB, myopia, ON-bipolar cells, GPR179, refractometry, dopamine

## Abstract

Mutations in *GPR179* are one of the most common causes of autosomal recessive complete congenital stationary night blindness (cCSNB). This retinal disease is characterized in patients by impaired dim and night vision, associated with other ocular symptoms, including high myopia. cCSNB is caused by a complete loss of signal transmission from photoreceptors to ON-bipolar cells. In this study, we hypothesized that the lack of *Gpr179* and the subsequent impaired ON-pathway could lead to myopic features in a mouse model of cCSNB. Using ultra performance liquid chromatography, we show that adult *Gpr179*^−/−^ mice have a significant decrease in both retinal dopamine and 3,4-dihydroxyphenylacetic acid, compared to *Gpr179*^+/+^ mice. This alteration of the dopaminergic system is thought to be correlated with an increased susceptibility to lens-induced myopia but does not affect the natural refractive development. Altogether, our data added a novel myopia model, which could be used to identify therapeutic interventions.

## 1. Introduction

Myopia is the most common ocular dysfunction worldwide with an alarming rise in prevalence, especially in Southeast Asia [1,2,3,4]. It is characterized by an abnormal increase in the axial length of the eye during refractive development (i.e., emmetropization process), leading to a disturbed far-sight but normal near-sight [1,2,5]. Previous studies of myopic patients and animal models have expanded the knowledge of mechanisms implicated in the onset and progression of myopia, but many cellular and molecular processes are still poorly understood. Myopia can be divided according to the refractive error of a patient between low myopia (≤−0.5 D) and high myopia (≤−6.00 D). High myopia can be a cause of blindness and lead to other ocular diseases such as glaucoma, retinal detachment and cataract [6,7]. Many studies have shown that exposure to outdoor light inhibits myopia onset and progression in both humans [7,8,9,10,11,12,13,14] and experimental animal models [15,16,17,18,19,20]. To date, many models from different species have been proposed to study myopia. Although larger animal models more similar to humans—such as monkeys and tree shrews—would be of interest, technical and ethical issues hinder their development and use. In contrast, the generation of mouse models is manageable but displays pros and cons. On the one hand, both the genetics and environment can be controlled and modified. On the other hand, the small size of the murine eye leads to difficulties in manipulation and interpretation of results during intraocular injection and ocular measurements [21,22,23,24], and to difficulties defining an end point for the emmetropization process. It was shown that one diopter change in refractive error corresponds to an increase of 5.4–6.5 μm in axial length of C57BL/6 mouse eyes [25,26] as opposed to the 280 µm to 400 µm changes in human children and adults, respectively [27,28]. Therefore, in small animals, it is necessary to induce myopia in order to measure detectable refractive errors until a technology overcomes this issue. Myopia induction can be performed using lens-induced myopia (LIM) or form deprivation myopia (FDM) protocols, both of which have been validated in mice and chickens [22,29,30,31,32,33,34].

Although the precise molecular mechanisms of refractive error onset and inhibition remain to be elucidated, previous work has highlighted the impact of the retinal content of the neurotransmitter dopamine (DA) and its receptors upon refractive development and myopia [35,36,37,38,39,40]. Indeed, in the retina, DA is synthesized by dopaminergic amacrine cells (DACs), its secretion is regulated by light conditions [41,42,43,44] and it requires a functional ON-pathway [29,45]. The retinal level of the DA degradation metabolite, 3,4 dihydroxyphenylacetic acid (DOPAC) is a commonly used marker of DA release and the DOPAC/DA ratio is considered a marker of DA turnover. It was shown that DA and DOPAC levels are reduced in a variety of myopia experimental models including FDM and LIM [17,43,44,46,47,48] and experimentally increased retinal DA levels inhibit the development of FDM and LIM in amphibians [42], mammals [43,44] and avian [48,49], and vice versa [38,40].

Several inherited retinal diseases are associated with high myopia (syndromic myopia) [7] including cCSNB [50] where patients display high myopia with a median error of −7.4 D [50,51]. cCSNB is a group of genetically heterogeneous and non-progressive retinal disorders. Its main clinical features consist of an impairment of dim light vision, often associated with other ocular signs such as high myopia, nystagmus and/or strabismus [50,51,52,53,54,55,56]. cCSNB is characterized by a normal a-wave but a severe or complete loss of the b-wave under scotopic conditions and altered b-waves under photopic conditions, as measured with electroretinography. Such abnormal electroretinograms are directly caused by the dysfunction of the ON-BC pathway [50,52,56,57]. To date, mutations causing cCSNB in patients were found in *NYX*, *TRPM1*, *GRM6*, *GPR179* and *LRIT3* genes [50,58,59,60,61,62,63,64,65,66,67]. Animal models mimicking those mutations already exist and have been extensively characterized: *Nyx* (also known as *nob* (no b-wave)) *Grm6*, *Trpm1, Gpr179* or *Lrit3* [29,62,68,69,70,71,72,73,74,75,76,77,78,79,80,81,82,83,84,85,86]. Previous studies have shown that *nob* mice (carrying a null mutation in *Nyx*, which causes a loss of function of nyctalopin) display a higher susceptibility to FDM than wild-type littermates. The retinas of ungoggled *nob* mice also have a lower level of light-induced DOPAC and an overall decrease in both DOPAC and DA levels compared with wild-type mice [29]. Similar results were found in mice lacking *Grm6* [45]. Interestingly, a mouse model with cone dysfunction (*Gnat2^−/−^*) also displayed a higher susceptibility to FDM, but the levels of retinal DOPAC or DA were unchanged [87]. Despite the fact that all the mutated genes in cCSNBs seem to affect the same pathway, and that patients and all animal models display ON-bipolar cell (ON-BC) transmission defects as shown by electroretinography (for review, see [50]), some differences have been noticed. *Lrit3*^−/−^ mice exhibit a slight decrease in thickness of the inner nuclear layer [77], while such an observation was absent in retinas of *Gpr179*^−/−^ mice [74].

All proteins encoded by cCSNB-related genes are involved in synaptic transmission between photoreceptors and ON-BCs [50]. In this study, we decided to focus our experiments on the *Gpr179^−/−^* cCSNB mouse model [74]. GPR179 is an orphan G-protein coupled receptor located at the dendritic tips of ON-BCs [75]. It was shown to interact with the presynaptic transmembrane protein pikachurin and the post synaptic RGS11/9 complex to modulate the activation of TRPM1 channel which is the final actor in the mGluR6-mediated cascade leading to the depolarization of ON-BCs [50,82].

In the mammalian retina, detection of light increments and decrements is mediated by the ON and OFF pathways, respectively. In a functional ON pathway, ON-BCs convey the visual information from photoreceptors to ON-retinal ganglion cells (ON-RGCs) in two different ways according to the origin of their input. ON-cone bipolar cells (ON-CBCs) receive direct input from cone photoreceptors and synapse onto ON-RGCs. At low light intensity, rod-bipolar cells (RBCs) receive their input from rods and synapse onto AII amacrine cells (AII-ACs) which, in turn, activate ON-CBCs via gap junctions and inhibit OFF-cone bipolar cells (OFF-CBCs) through inhibitory synapses. ON-CBCs then synapse onto ON-RGCs. In a functional OFF pathway, OFF-CBCs convey the visual information from cone photoreceptors (but also from a minority of rods) to OFF-retinal ganglion cells (OFF-RGCs). Ultimately, RGCs indirectly drive the visual information to the visual cortex through the optic nerve and thalamus [88,89]. In addition to these signaling pathways, the activity of ACs is noteworthy since DACs receive input from AII-ACs and CBCs and some subtypes of most major classes of retinal neurons harbor dopamine receptors (D1R to D4R) [90]. As functional GPR179 is necessary for a proper ON signaling, and as DACs receive indirect excitatory input from ON-BCs, one might expect that the loss of GPR179 could cause a reduced activity of DACs and thus, a reduction in DA release.

The aim of our study was to monitor whether the lack of GPR179 in a novel model of cCSNB [74] has an impact upon DA turnover and induced-myopia. We measured the retinal DA and DOPAC levels, the natural refractive development and the sensitivity to LIM induction in *Gpr179^−/−^* mice in order to improve our understanding of the role of cCSNB molecules and the ON signaling cascade upon retinal function and myopia onset and progression.

## 2. Results

### 2.1. Measurement of Retinal Levels of DA and DOPAC

To determine whether the loss of *Gpr179* induces a change in dopamine metabolism and activity, we quantified the levels of DA and DOPAC in the retinas of adult (post-natal day 80, P80) *Gpr179^+/+^* and *Gpr179^−/−^* mice after 4 hours of light adaptation (Figure 1) at 50 lux. *Gpr179^−/−^* retinas display a significant decrease in DA (Figure 1A) compared with wild-type littermates. Mean DA levels were 3022 ± 214 fmoles/retina in *Gpr179^+/+^* compared with 1310 ± 135 fmoles/retina in *Gpr179^−/−^*. Similarly, *Gpr179^−/−^* retinas display a significant decrease in DOPAC (Figure 1B) compared with wild-type littermates. Mean DOPAC levels were 844 ± 38 fmoles/retinas in *Gpr179^+/+^* compared with 413 ± 25 fmoles/retinas. No difference in the DOPAC/DA ratio was observed in *Gpr179*^−/−^ compared to *Gpr179*^+/+^ (Figure 1C). These results indicate that mice lacking *Gpr179* exhibit disturbances in dopamine metabolism.

### 2.2. Kinetics of Refractive Development

In order to establish whether the loss of *Gpr179* and the resulting dysfunctional ON-BC pathway could lead to change in natural refractive development, we measured the refractive state of both eyes of *Gpr179^+/+^* and *Gpr179^−/−^* mice from 3 to 9 weeks of age (Figure 2). In both genotypes tested, we observed a hyperopic shift from 3- to 6-weeks-old, starting from 0 to +1 D at 3-weeks-old and reaching +3 D to +4 D at 6-weeks-old. From 6- to 9-weeks-old a plateau was reached at +3 to +4 D for both genotypes. Overall, no significant change in the refractive state was observed between *Gpr179^−/−^* mice and their wild-type littermates, therefore loss of GPR179 did not induce any measurable change of refractive development.

### 2.3. Assessment of Sensitivity to Myopia Induction

To test whether the loss of GPR179 could affect the susceptibility to environmentally-induced myopia, we generated a LIM protocol in *Gpr179^+/+^* and *Gpr179^−/−^* mice from P21 to P42 and measured the mean interocular shift (called myopic shift when negative) at different time points: P21 (Figure 3; 0-week post-goggling), P35 (2 weeks post-goggling) and P42 (3 weeks post-goggling). At 0-week, no myopic shift was observed in *Gpr179^+/+^* nor in *Gpr179^−/−^* mice and no difference of myopic shift between the two genotypes was measured. At 2 weeks post-goggling, we recorded a significant difference in the myopic shift between *Gpr179^+/+^* and *Gpr179^−/−^* mice. Myopic shift reached −7.0 ± 0.9 D in *Gpr179^+/+^* and −10.9 ± 0.8 D in *Gpr179^−/−^* mice (*p* < 0.01). At 3 weeks post-goggling, the difference in the myopic shift between *Gpr179^+/+^* and *Gpr179^−/−^* mice increased compared to week 2. The myopic shift in *Gpr179^+/+^* mice slightly increased to −8.3 ± 0.6 D while it reached −16.1 ± 0.8 D in *Gpr179^−/−^* mice (*p* < 0.001). Overall, these data demonstrate an impact of the loss of GPR179 upon the sensitivity to lens-induced myopia.

## 3. Discussion

For the last two decades, the impact of ON-pathway defects upon refractive development and myopia onset has raised attention. Although the detailed mechanisms of normal and myopic refractive development are still unclear, the development of new induction protocols and the extensive use of transgenic models, that allow for the manipulation of both environmental and genetic factors, help resolve the signaling cascades involved. To date, many factors have been proposed to trigger or stop the growth of the eyeball and thus, to impact refractive development [39,43,91,92,93,94,95,96]. Among them, the dopaminergic system is the most characterized [17,47,90,97]. In this study, we report for the first time the presence of myopic features in the *Gpr179* mouse model [74].

Our experiments revealed drastic decreases of both DA and DOPAC levels in the light-adapted retinas of *Gpr179^−/−^* mice compared to their wild-type littermates (Figure 1). DA is reduced in *Gpr179*^−/−^ mice by two-thirds while DOPAC is halved. Functional GPR179 is necessary for a normal ON signaling, and DACs receive indirect excitatory input from ON-BCs. Furthermore, stimulation of the ON pathway induces the release of DA via the activation of DACs [42,90,98]. Thus, it has been hypothesized that defects in ON-BCs signaling could impact DA release [29,45,99]. Our results are in line with previous studies addressing this hypothesis [29,45] Still, some differences with previous findings are noteworthy. Here, both DA and DOPAC levels are reduced in *Gpr179^−/−^* mice while only DOPAC was found to be decreased in *Grm6*^−/−^ mouse retinas [45]. In the retinas of *nob* mice, both DA and DOPAC levels were lower than in wild-type mice [29]. A disparity in tested ages was proposed to explain some of the observed differences as an impact of age upon retinal DA and DOPAC levels has been reported [100]. Our results are consistent with this explanation since both *Gpr179^−/−^* and *nob* [70] mice were 12 weeks old but mice lacking *Grm6* [45] were 16 weeks old. Another explanation could involve the lighting conditions used for light adaptation [17] which may have been different in the previous studies [29,45] compared to ours. A more interesting explanation might be that although both GRM6 and GPR179 are necessary for ON-BC signaling and are thought to be expressed in all ON-BCs and similarly localized to their dendritic tips, it is not known if the physiological consequences of their absence are the same. Indeed, the light responses of retinal ganglion cells (downstream of ON-BCs) are different in mice with different *Grm6* mutant alleles [83], thus it is possible that DACs are differentially affected in *Grm6^−/−^* and *Gpr179^−/−^* mice.

Our light adaptation was mesopic. Thus, one can hypothesize that residual ON or OFF activation could affect DA metabolism since DACs receive inputs from ON and, to a lesser extent, from OFF pathways [90]. This hypothesis seems unlikely as (1) all the cCSNB mouse lines tested display a similar loss of ON-pathway function [70,74,86]. (2) To our knowledge OFF responses have not been characterized in the *Gpr179^−/−^* mouse line, but other studies show that the OFF pathway is globally unaltered in cCSNB patients [50,101] and mouse models [77,102]. (3) Experiments performed in a *Vsx1*^−/−^ mouse, a model with dysfunctional OFF bipolar cells [103], revealed no impact of the OFF pathway defect upon refractive development and myopia onset [104] nor on dopamine metabolism in adult *Vsx1*^−/−^ retinas [99].

The DOPAC/DA ratio is unchanged in *Gpr179*^−/−^ compared to *Gpr179*^+/+^ (Figure 1C). Several studies have noted the impact of an altered DOPAC/DA ratio upon sensitivity to myopia induction [29,43,45,105]. In our study, both DOPAC and DA levels are reduced in *Gpr179*^−/−^ mice by similar levels, leading to an unchanged DOPAC/DA ratio. Overall, our results are in line with the hypothesis of a defective ON-pathway causing a reduction of retinal dopamine.

Strikingly, comparing the natural refractive development of *Gpr179*^+/+^ and *Gpr179*^−/−^ mice, we found no difference in the refractive state between the two genotypes across age. Previous studies performed on other cCSNB models display variable results: *Nyx*^−/−^ mice become more hyperopic than wild-type littermates [29], while *Grm6*^−/−^ mice display a more myopic development [45]. Furthermore, both *Gpr179*^+/+^ and *Gpr179*^−/−^ mice are more myopic in general than some genetic mouse models [29,30,45], but similar to others [17,104,106,107]. Both genetic [104,108] and environmental [17,107] factors can influence refractive development. We used the same mouse background strain, C57BL/6J as others [29,45,104,108], but as mentioned above, lighting conditions can differ between laboratories. One study also found an impact of circadian rhythm upon axial length [109], with a range of 20 µm difference between the maximum and the minimum, which would cause a theoretical change in refractive state of about 4–5 D. In the present study, we did not consider this potential cause of variability. Another cause of disparity could be the use of eye drops and the absence of anesthetics in our experimental paradigm. It was shown that intra-peritoneal injection of ketamine/xylazine could induce a hyperopic shift in mice [110]. We tested the use of isoflurane inhalation instead but found the same issue as with ketamine/xylazine anesthesia (data not shown). However, the time course of refractive development follows the same pattern as in other studies: a hyperopic shift followed by a plateau reached at 6 weeks of age [29,45,104]. Many studies have noted the difficulty of measuring myopia in small animals because of the small eye artifact [29,45,111]. Our data confirm that measurement of myopia in small animals often requires an induction protocol.

By developing a 3-week-long lens-induced myopia protocol, we found a substantial increase of the myopic shift in *Gpr179*^−/−^ mice compared to wild-type littermates (Figure 3). Consistent with previous findings with cCSNB models, this myopic shift is significant at 2 weeks post-goggling [29,45], but continues to increase until 3 weeks. Surprisingly, this myopic shift is also much larger in *Gpr179*^+/+^ mice than in the control mice used in those previous studies. As mentioned before, we used the same mouse strain, but the lightning environment might have had an impact. However, the most relevant cause of difference could be found in the induction protocol. We used a −25 D lens as opposed to monocular deprivation and started the induction one week earlier than in the studies mentioned above. Our results are in line with previous studies from other groups showing that starting the induction at P21 [46] or P24 [112] instead of P28 causes a more substantial increase in myopic shift.

We need to elucidate whether the myopic shift induced by our LIM protocol directly leads to a decrease in retinal levels of DA and/or DOPAC in wild-type and in *Gpr179^−/−^* mice. Interestingly, recent studies showed that injecting DA agonists suppresses FDM in several animal models [48,113,114]. In addition, induction of FDM seemed to decrease retinal levels of DA in guinea pigs [115], chickens [116] and primates [91] but not in wild-type mice [117]. This may be at least partly explained by the fact that most mice used in myopia studies are melatonin-deficient mice (i.e., C57BL/6J mice). Melatonin is released by photoreceptors, is implicated in circadian rhythm, and seems to inhibit dopamine release. Dopamine and melatonin act in opposite ways, interacting with each other to form an inhibitory feedback loop [118]. In support of the notion that melatonin is required for FDM-related retinal DA changes, a recent study showed that melatonin-proficient mice (i.e., CBA/CaJ) display reduced retinal levels of DA during FDM [119]. Similarly, this may explain a part of the residual dopamine release in the retinas of cCSNB mouse models observed in our study and by others [29,45]. Further experiments are required to determine whether the causal link between DA metabolism and sensitivity to myopia induction is unidirectional or bidirectional in mice. Mechanisms implicated in LIM-mediated myopic shift seem to be less dependent on DA metabolism than those mediated by FDM [43, 47], but are clearly not independent either. This can be explained by the loss of the timing cue of lighting occurring during FDM but not during LIM [47].

It is noteworthy that injecting L-AP4 in chicken eyes causes a hyperopic shift in ungoggled eyes but does not have an impact on form-deprived ones [120,121]. L-AP4 is an agonist of group III metabotropic glutamate receptors. In the retina, it stimulates mGluR6 and whether it causes the ON-BCs to be constitutively hyperpolarized remains unknown. To our knowledge, there is still little evidence on the resting state of ON-BCs in cCSNB models. Whole-cell patch-clamp recordings of RBCs from *nob* mice retinal slices revealed small holding currents, indicating that channels were mostly closed or absent [122]. Further studies focusing on the differential effect of L-AP4 upon ungoggled eyes and form-deprived ones might be of interest to decipher the mechanisms leading to myopia in cCSNBs.

Altogether, our results indicate that the loss of GPR179 induces disturbances in the dopaminergic system. The ON-pathway defect due to the lack of GPR179 does not have any measurable impact upon refractive development but increases the sensitivity to LIM. Overall, our data strengthen the hypothesis that ON-pathway deficiency can be associated with myopia onset through alteration of dopaminergic signaling. This work validates the potential use of cCSNB models to study myopia. Thus, we think they can be used to identify new genes related to myopia in cCSNB. For instance, by performing RNAseq studies upon cCSNB models and patients, we may find genes implicated in syndromic and/or common myopia.

## 4. Material and Methods

### 4.1. Animal Care and Ethical Statement

All animal procedures were performed according to the Council Directive 2010/63EU of the European Parliament and the Council of 22 September 2010, on the protection of animals used for scientific purposes, with the National Institutes of Health guidelines and with the ARVO Statement for the Use of Animals in Ophthalmic and Vision Research. They were approved by the French Minister of National Education, Superior Education and Research. Mouse lines and projects were registered as following: refractometry: APAFIS #27474 2020100110251857 v5. Mice were kept in 12:12 hour light:dark cycles with mouse chow and water ad libitum.

### 4.2. Polymerase Chain Reaction (PCR) Genotyping

DNA was extracted from mouse tails with 50 mM NaOH after incubation at 95 °C for 30 min. Wild-type and mutant allele were amplified independently using a polymerase (HOT FIREPol, Solis Biodyne, Tartu, Estonia), a common forward primer: mGpr179_EF (5′-CTGCCCCCACAGAATGTTCCCA-3′) and two specific reverse primers: mGpr179_Er2 (5′-CACCGCCTCTTTACTCTGCCCA-3′) for the wild-type allele and mGpr179_Kr (5′-GGGCAAGAACATAAAGTGACCCTCC-3′) for the mutant allele. The following PCR program was used: 15 min at 95 °C for denaturation, 35 cycles of 45 sec at 95 °C, 1 min at 60 °C, and 1.3 min at 72 °C. A final extension for 10 min at 72 °C was performed. This generates the following amplicons: PCR using mGpr179_EF and mGpr179_Er2 primers amplifies a product of 146 base pairs (bp) for the wild-type allele and no product for the mutant allele, PCR using mGpr179_EF and mGpr179_Kr primers amplifies no product for the wild-type allele and a 303 bp product for the mutant allele. PCR products were separated by electrophoresis on 2% agarose gels, stained with ethidium bromide, and visualized using a documentation system (Molecular Imager^®^ Gel Doc^TM^ XR+ System, Bio-Rad, Hercules, CA, USA).

### 4.3. DA and DOPAC Measurements

After 4 hours of light adaptation in a light controlled room at 50 lux, mice were sacrificed at 12 pm by CO_2_ inhalation followed by cervical dislocation. Retinas were isolated, frozen in liquid nitrogen and stored at −80 °C. Amounts of DA and DOPAC were quantified with ultra performance liquid chromatography (UPLC) with coulometric detection adapted from [123]. The UPLC equipment consisted of a system (Ultimate 3000, Thermo Fisher Waltham, MA, USA) in isocartic mode with two working electrodes (Coulochem III, Thermo Fisher, Waltham, MA, USA). Analytes were separated on a C18 column (150 mm × 4.6 mm, 3 µm particle size, BDS Hypersil, Thermo Fisher, Waltham, MA, USA), maintained at 40 °C, with a mobile phase (Thermo Fisher Test). Briefly, tissue samples were crushed for 30 s (Ultraturax, IKA, Staufen, Germany) in 0.2 M perchloric acid containing EDTA and ascorbic acid (0.2 µM each). After centrifugation at 25,000x g for 30 min at 4 °C, the supernatants were passed through a 10 kD cutoff filter (Nanosep, Port Washington, NY, USA) at 12,000 g, for 10 mins at 4 °C. Finally, 50 µL of the ultrafiltrates were injected. The two serial coulometric electrodes with applied potentials −100 and 350 mV were used for the measurements of DA and DOPAC.

Statistical analyses were performed using Prism 9.1.2, GraphPad (GraphPad v7 Software Inc., La Jolla, CA, USA). Statistical significance was measured with a non-parametric two-tailed Mann–Whitney test.

### 4.4. Lens-Induced Myopia

To induce myopia, P21 mice were anesthetized by isoflurane inhalation (5% induction, 2% maintenance). The scalp was cut to expose 1.2 cm of the skull in the rostrocaudal axis. Two intracranial screws were implanted on both left and right sides of the skull at y = −2 mm from the bregma. A homemade goggle frame was placed on the skull and fixed using dental cement (FujiCEM™, Phymep^©^ Cat #900903, Paris, France). This goggle frame, adapted from X. Jiang et al.; 2018 [46], was built in resin using a 3D printer. The −25 D lenses were stuck on the frame using glue (vetbond™, Phymep^©^, Paris, France). Stitches were used to avoid displacement of lens by mice. The −25 D lens was always placed in front of the right eye.

### 4.5. Refractometry

After a dark adaptation for 30 min, eye drops were used to dilate the pupils: 0.5% mydriaticum^©^ (Théa, Clermont Ferrand, France), 5% neosynephrine (Europhta, Monaco). The mice were placed on a homemade restraining platform in front of an eccentric infrared photorefractometer calibrated as described previously [32]. Calibration was verified using lenses of increasing power, from −10 D to +10 D placed in front of a mouse eye. The mouse was positioned so that the first Purkinje image was in the center of the pupil. The data were then recorded using software designed by F. Schaeffel [32]. We collected two data sets per animal, one per eye, each data set consisted of 90 measures, each of which was the mean of 10 successful measurements (in total: 1800 points/animal). For refractive development experiments, the measurements were performed once per week, every week from 3- to 9-weeks-old and the mean refractive state of both eyes was used for statistical analysis. To evaluate the sensitivity to myopia induction, the difference between goggled and ungoggled eye (referred to as interocular shift) measured at post-natal day 21 (P21, day of surgical procedure), P35 (14 days after surgery) and P42 (21 days after surgery) was used for statistical analysis. Only the mice displaying less than 2 D of interocular shift at P21 were used for the induction protocol. Statistical analyses were performed using Prism 9.1.2, GraphPad (GraphPad v7 Software Inc., La Jolla, CA, USA). Statistical significance was measured with a two-way ANOVA test.

## Figures and Tables

**Figure 1 ijms-24-00219-f001:**
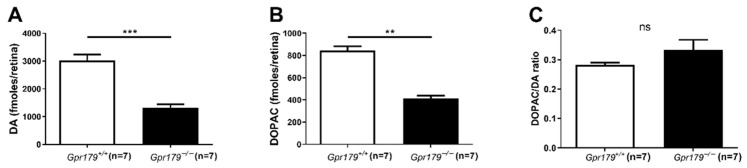
Retinal levels of DA and DOPAC. Quantification of DA (**A**), DOPAC (**B**) and the DOPAC/DA ratio (**C**) in light-adapted adult *Gpr179*^+/+^ and *Gpr179*^−/−^ retinas. Data are expressed as Mean ± SEM. ns: not significant, **: *p* ≤ 0.01, ***: *p* < 0.001.

**Figure 2 ijms-24-00219-f002:**
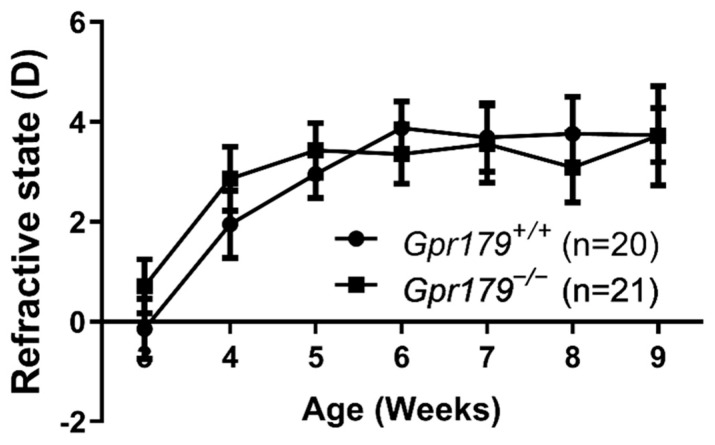
Natural refractive development. Measurement of the refractive state of *Gpr179*^+/+^ and *Gpr179*^−/−^ mice across age. Data are expressed as Mean ± SEM.

**Figure 3 ijms-24-00219-f003:**
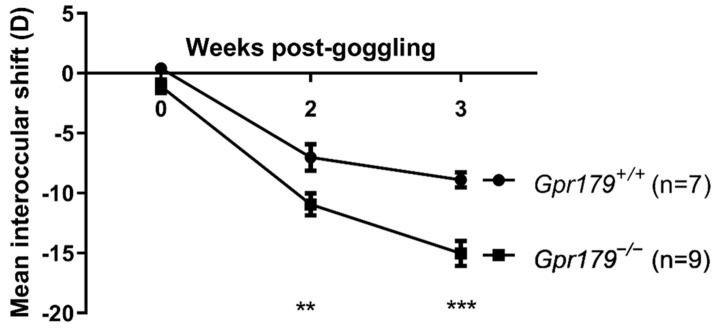
Lens-induction of myopia. Measurement of the kinetics of mean interocular shift between the goggled and ungoggled eyes of *Gpr179^+/+^* and *Gpr179^−/−^* mice. Data are expressed as Mean ± SEM. **: *p* ≤ 0.01, ***: *p* < 0.001.

## Data Availability

Non-applicable.

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
