# Peer review of "Mice Lacking Gpr179 with Complete Congenital Stationary Night Blindness Are a Good Model for Myopia"

_ijms, 2022, doi:10.3390/ijms24010219_

Round 1
Reviewer 1 Report
The manuscript includes interesting observation and experimental data reagarding myopia and an IRD-associated XL-gene GPR179. The authors succeeded to present that GRP179 abnormality leads to myopia in mice, which is one of the famous complications in XL-IRD. Discussion of the manuscript is informative.
Author Response
Dear Dr. Spasojevic, dear reviewers,
We are grateful for the invitation to submit a revised version of our manuscript “Mice lacking Gpr179 with complete congenital stationary night blindness are a good model for myopia”, ID Ref.: ijms-1980861. We are also thankful for the fruitful comments, which helped us improve the quality of the manuscript. Enclosed we return the revised version. We have responded to all criticisms and suggestions and we list below how and where we have implemented the changes to the new version of the manuscript. In addition, all other changes are visualized by “track-changes”. Thank you for your patience!
Yours sincerely,
Baptiste Wilmet and Christina Zeitz
Reviewer 1
Comments and Suggestions for Authors
The manuscript includes interesting observation and experimental data regarding myopia and an IRD-associated XL-gene GPR179. The authors succeeded to present that GRP179 abnormality leads to myopia in mice, which is one of the famous complications in XL-IRD. Discussion of the manuscript is informative.
We are thankful for the positive comment of the reviewer. Just to clarify, the GPR179 gene is located on chromosome 17q12 and not on the X-chromosome. However, we feel this does not need to be mentioned in this article since we are not talking about myopia in X-linked inherited retinal disorders, but about myopia in complete congenital stationary night blindness. Line 94: “Several inherited retinal diseases are associated with high myopia (syndromic myopia) [7] including cCSNB [53] where patients display high myopia with a median error of -7.4 D [53, 54]”
Reviewer 2 Report
Please see separate attachment.

Author Response
Dear Dr. Spasojevic, dear reviewers,
We are grateful for the invitation to submit a revised version of our manuscript “Mice lacking Gpr179 with complete congenital stationary night blindness are a good model for myopia”, ID Ref.: ijms-1980861. We are also thankful for the fruitful comments, which helped us improve the quality of the manuscript. Enclosed we return the revised version. We have responded to all criticisms and suggestions and we list below how and where we have implemented the changes to the new version of the manuscript. In addition, all other changes are visualized by “track-changes”. Thank you for your patience!
Yours sincerely,
Baptiste Wilmet and Christina Zeitz
Reviewer 2
IJMS 1980861 – Myopia in GPR179-/- mouse model cCSNB – Review To the Authors:
General Comments: This is an interesting and informative follow-up to previously published studies, involving some of the same authors, on the GPR179 gene and its role in CSNB1. Although I’m not very fond of the mouse as a model for myopia studies – because its vision is so much adapted to low light intensities (rod vision), and because its eye is so very small and has great depth of focus – it’s undeniably powerful and convenient for some purposes because of its superior potential for genetic manipulation. A consistent (though not universal) feature of human CSNB1 is the development of moderate to high myopia. Thus the rationale of using a mouse knockout for this gene for studying the retinal mechanisms that are involved in emmetropization and myopization is well founded.
Strengths: The aims are simple: to test the effect of GPR179-KO on normal refractive development and the susceptibility to lens-induced myopia, and to determine whether the deletion affected the metabolism of dopamine (a common factor in cause/prevention of myopia in animal models). The availability of mice having GPR179 gene knockout is a major advantage. The experimental design is straightforward, and the procedures – autorefraction in small animals, and measurement of retinal dopamine (DA) and its post-release metabolite, DOPAC – are well established and proven in the authors’ labs. The results are trustworthy, and they add another tool and mammalian model to aid in understanding the retinal mechanisms in myopia. Most importantly, the results further support the involvement of retinal ON-pathways – specifically, those involving depolarizing bipolar cells (ON-BCs) – in emmetropization and myopization.
We thank the reviewer for the positive comment. We agree with the statements about the pros and cons of the use of mice model in myopia research.
Weaknesses: No explicit hypothesis is stated. Paraphrasing the statement in the Abstract, I suppose it would be something like, “lack of Gpr179 and consequent impairment of ON-pathways leads to myopic features in a mouse model of cCSNB.” My problem with this is that the work doesn’t really test such a hypothesis; that is, it describes that the mutant mice show some (but not all) of the characteristics that are expected from defects in ON-pathways, but it does not test whether those changes cause myopia. Indeed, the mutant mice did not develop myopia, unless they were challenged by imposed hyperopic defocus (minus-lens treatment); and there is not a clear, unambiguous path from the mixed results of DA and DOPAC measurements, to the conclusion that “This alteration of the dopaminergic system leads to an increased susceptibility to lens-induced-myopia.” It is clearly associated with such increase in susceptibility, but the authors have not tested whether it is the cause – and if they had done so, they would have added little to what is already known from the long history of animal studies linking the action(s) of DA to the cause and prevention of myopia.
We thank the reviewer for pinpointing the lack of a clear hypothesis and the subsequent exaggerated conclusions. About the hypothesis, changes have been made in the abstract to highlight it: abstract line 27: “We hypothesized that the lack of Gpr179 and the subsequent impaired ON-pathway could lead to myopic features in a mouse model of cCSNB.”
We agree that we did not perform experiments showing a causal link between ON pathway defect and alterations in DA metabolism, nor between DA metabolism and myopia onset in mice. Since the causal link between alterations in the retinal dopaminergic pathway and the onset of myopia in both avians and mammals were made by a large amount of evidence in the literature, we cited many references in the introduction of the current version of the manuscript, lines 66-78.
In guinea pigs: Jiang L, Long K, Schaeffel F, et al. Effects of dopaminergic agents on progression of naturally occurring myopia in albino guinea pigs (Cavia porcellus). Invest Ophthalmol Vis Sci.2014;55:7508–7519. DOI:10.1167/ iovs.14-14294.
In mice: Chen S, Zhi Z, Ruan Q, et al. Bright light suppresses form-deprivation myopia development with activation of dopamine D1 receptor signaling in the ON pathway in retina. Invest Ophthalmol Vis Sci. 2017;58:2306–2316. DOI:10.1167/ iovs.16-20402.
In primates: Iuvone PM, Tigges M, Stone RA, Lambert S, Laties AM. Effects of apomorphine, a dopamine receptor agonist, on ocular refraction and axial elongation in a primate model of myopia. Invest Ophthalmol Vis Sci 1991; 32: 1674–1677.
In chickens: Li, X. X.; Schaeffel, F.; Kohler, K.; Zrenner, E., Dose-dependent effects of 6-hydroxy dopamine on deprivation myopia, electroretinograms, and dopaminergic amacrine cells in chickens. Visual neuroscience 1992, 9, (5), 483-92.
Previous and our studies only indirectly showed the impact of an ON-pathway defect upon DA metabolism. Therefore, we softened our statement in the abstract to: “This alteration of the dopaminergic system is thought to be correlated with an increased susceptibility to lens-induced-myopia but does not affect the natural refractive development.” Lines 31-33.
Specific Comments [line number in brackets]: [Terminology]: My geneticist colleagues tell me that (a) while the names of human genes and mRNA are conventionally presented as all upper-case (e.g., GPR179), those of mice are lowercase except for the initial letter (e.g., Gpr179); and (b) the name of the gene, or its mRNA, should be in italics, but that of the protein, in plain text (e.g., in mouse, Gpr179 gene or mRNA, and Gpr179 protein). I haven’t attempted to make these distinctions in my review, but probably you should do so in making revisions.
We agree with most of the comments of the reviewer: According to HUGO and MGCN, human genes including mRNAs are written in upper case and italics, those of mice are in lower case except for the initial letter and italics However, all proteins (human and mice) are written in upper case and non-italic. We re-checked our manuscript and made the changes, when necessary.
[23] The beginning of the Abstract gets into the topic without telling the reader what GPR179 is; some explanation is needed here as to what this gene encodes. Perhaps a paraphrase of Ray et al, 2 ARVO 2012: "GPR179 is an orphan G protein-coupled receptor that interacts with GRM6 and is part of a protein network responsible for DBC signal transduction." (I also like very much the simple schematic in Ref#84, Orlandi et al., 2018, and perhaps an abbreviated description from that work would serve well here).
We thank the editor for pinpointing the lack of a description of the function of GPR179. We added a paragraph in the introduction summarizing the known function of GPR179 in mouse: lines 119-125: “All proteins encoded by cCSNB-related genes are involved in synaptic transmission between photoreceptors and ON-BCs [53]. In this study, we decided to focus our experiments on the Gpr179-/- cCSNB mouse model [77]. GPR179 is an orphan G-protein coupled receptor located at the dendritic tips of ON-BCs [78]. It was shown to interact with the presynaptic transmembrane protein pikachurin and the post synaptic RGS11/9 complex to modulate the activation of Trpm1 channel which is the final actor in the mGluR6-mediated cascade leading to the depolarization of ON-BCs [53, 85]”
[24] I think that “CSNB1” is generally preferred over “cCSNB”; no?
Originally“CSNB1” refers to the genetic locus implicated in cCSNB (NYX gene defect). We prefer to use the term complete congenital stationary night blindness, which describes the clinical and pathophysiological components of the disorder.
[25] "dim and night vision": (a) "dim" is a subjective term – better to use "low-intensity"; and (b) “night vision” ties function to time of day, rather than lighting – so I suggest changing to " "night blindness" - impaired vision in low intensity light".
We thank the reviewer for the precision. Terms “dim and night vision” are relevant terms when it comes to clinical description of patients, such as in the abstract. But we agree it is irrelevant and lacking accuracy when it comes to description of the lightning conditions of mice or the impact of lightning conditions upon retinal function.
The term “night vision” was used only to describe clinical features of cCSNB and the terms “dim light” and “dim vision” were changed in the introduction: line 131: “In dim light, rod-bipolar cells (RBCs) receive their input from rods and synapse onto AII Amacrine cells (A2-ACs) which, in turn etc…” Was replaced by “at low light intensity, rod-bipolar cells (RBCs) receive their input from rods and synapse onto A2 Amacrine cells (A2-ACs) which, in turn etc…”
[29] Insert “retinal content” (of dopamine ... )
We thank the reviewer for the precision. The required change was made as proposed.
[30-31] “...alteration ... leads to an increased susceptibility ... “: Bear in mind that this alteration could be due to a number of changes – some of which (such as reduced rate of DA synthesis) would not preclude still-significant levels of DA release, at least in the short term. In my opinion, the work would be greatly strengthened by measuring the rate of DA synthesis, which is the main determinant of dopaminergic activity; alternatively, measuring DA and DOPAC content in the vitreous would provide a widely accepted measure of DA-release (yes, admittedly there’s not much vitreous in the mouse eye, but ... J ). ... *Finally, if retinal content of DA and DOPAC, and the retinal DOPAC:DA ratio, do indicate the levels of DA synthesis/release in the retina, you can strengthen your rationale by stating so and citing key references. Remember that this journal is intended for a rather general readership.
We agree with the reviewer and softened our assumption in the abstract to: lines 31-34: “This alteration of the dopaminergic system is thought to be correlated with an increased susceptibility to lens-induced-myopia but does not affect the natural refractive development.” We thank the reviewer for further research directions to link DA and the development of myopia, but think that additional experiments are beyond the scope of this paper.
[52] “lead to an eye size related challenge” is not very clear; perhaps something like, “present challenges for intraocular injection and ocular measurements due to the very small size of the eye".
We agree that the chosen term is blurry and are grateful for highlighting this point. The sentence was changed to: lines 54-56 “On the other hand, the small size of the murine eyes leads to difficulties in manipulation and interpretation of results during intraocular injection and ocular measurements [21-24], and to difficulties defining an end point for the emmetropization process”.
[61-63] “the impact of ... (DOPAC) upon refractive development and myopia”: I wasn't aware that DOPAC is anything other than a convenient experimental indicator of DA release – certainly not that it plays an active role in (has "impact" upon) these processes. Do you really mean this?
We thank the reviewer for highlighting this point. It is nothing but a misuse of language, we mixed the DA, DA-R and DOPAC terms as being parts of the same pathway. Retinal levels of DOPAC by themselves are not known to have an impact upon refractive development and emmetropisation, as opposed to DA release. The sentence was changed to make it clearer as follows: lines 66-74: “Although the precise molecular mechanisms of refractive error onset and inhibition remain to be elucidated, previous work has highlighted the impact of the retinal content of the neurotransmitter dopamine (DA) and its receptors upon refractive development and myopia [35-40]. Indeed, in the retina, DA is synthesized by dopaminergic amacrine cells (DACs), its secretion is regulated by light conditions [41-44] and requires a functional ON-pathway [29, 45]. The retinal level of the DA degradation metabolite, 3,4 dihydroxyphenylacetic acid (DOPAC) is a commonly used marker of DA release and the DOPAC/DA ratio is considered a marker of DA turnover”.
[66-67]: “increased retinal DA and/or DOPAC levels blocks FDM and LIM”: I think what you mean here is that increases in DA release or extracellular availability (not retinal content, strictly speaking – and not DOPAC, right?) inhibit FDM and LIM, while decreases promote FDM and LIM. It would be better to say it something like that way.
Consistent with the previous comment, we confounded here the use of DOPAC as a marker of the release of DA and the impact of DA release itself upon myopia onset. This sentence was changed as follow: lines 74-78: “It was shown that DA and DOPAC levels are reduced in a variety of myopia experimental models including FDM and LIM [17, 43, 44, 46-48] and experimentally increased retinal DA levels blocks FDM and LIM in amphibians [42], mammals [43, 44] and avian [48, 49], and vice versa [38, 40].”
[68] Somewhere – around here, perhaps – you need to add a few sentences explaining why inactivation of GPR179, causing impairment of ON-BC responses, is expected to alter DA metabolism.
We agree with the reviewer that this point needs to be highlighted. We decided to move the paragraph describing the mechanisms of ON/OFF pathways in the mammalian retina under the paragraph describing the function of GPR179, at the very end of the introduction. And we added the following sentence: lines 132-144: “At low light intensity, rod-bipolar cells (RBCs) receive their input from rods and synapse onto AII amacrine cells (AII-ACs) which, in turn, activate ON-CBCs via gap junctions and inhibit OFF-cone bipolar cells (OFF-CBCs) through inhibitory synapses. ON-CBCs then synapse onto ON-RGCs. In a functional OFF pathway, OFF-CBCs convey the visual information from cone photoreceptors (but also from a minority of rods) to OFF-retinal ganglion cells (OFF-RGCs). Ultimately, retinal ganglion cells indirectly drive the visual information to the visual cortex through the optic nerve and thalamus [50, 51]. In addition to these signaling pathways, the activity of ACs is noteworthy since DACs receive input from AII-ACs and CBCs and most major classes of retinal neurons harbor dopamine receptors (D1R to D4R) [52]. As functional GPR179 is necessary for a proper ON signaling, and as DACs receive indirect excitatory input from ON-BCs, one might expect that the loss of GPR179 could cause a reduced activity of DACs and thus, a reduction in DA release.”
[69], “perception”: To my way of thinking, this term refers to the subjective experience of seeing, which is not generated in the retina per se; presumably you mean something like "detection" or "signalling".
We agree with the reviewer and replaced the word “perception” by “detection” line 127.
[74] “A2-ACs”: Here, and subsequently, why don’t you use the generally adopted term, AII-AC?
We were afraid of a possible misunderstanding of the roman numeral II, for double l, all ACs. But this fear might be irrelevant, “A2-ACs” have been replaced by “AII-ACs”, as requested.
[82], “most retinal neurons harbor dopamine receptors”: I think that it would be impossible to prove this, except perhaps by scRNA-seq, and in any case I think that probably it is not true. To be on the safe side, try something like: "Most [or even “All”] major cell classes in the retina express at least one type of dopamine receptor".
We agree with the reviewer, indeed, we meant “most major retinal cell classes express at least one type of dopamine receptor”. The requested change was done line: 141.
[86-87] “main clinical features [of cCSNB] consist of an impairment of ... high myopia ... “: You should indicate that this is usually true, but not always. Not always; see, e.g.: Dry KL et al., Clin 3 Genet 1993, Linkage analysis in a family with complete type congenital stationary night blindness with and without myopia. doi: 10.1111/j.1399-0004.1993.tb03812.x. – “In two affected male cousins, one had congenital nystagmus and myopia, while the other was initially thought to have retinitis pigmentosa with optic atrophy and was hyperopic.”
We thank the reviewer for the reference and the precision. We added the reference and modified the sentence, lines: 96-99: “Its main clinical features consist of an impairment of dim light vision, often associated with other ocular symptoms such as high myopia, nystagmus and/or strabismus [53-59]”
[96] Insert: ... The “retinas of” ungoggled nob mice... (this should ne done in all similar situations)
We made the changes as requested.
[101-103] “all the mutated genes in cCSNBs are implicated in the same pathway ... patients and all animal models display ON-BC transmission defects ... “: This is likely true only in a trivial sense, I think; it’s possible that some cases of CSNB1 are due to mutations affecting primarily photoreceptor cells,, or interneurons (ACs) that are not directly in the ON-pathways. So, please consider stating this a bit more cautiously.
Lines 113-118:To soften this paragraph, we write in the current version of the manuscript: “Despite the fact that all the mutated genes in cCSNBs seem to affect the same pathway, and that patients and all animal models display ON-bipolar cell (ON-BC) transmission defects as shown by electroretinography (for review, see [2]), some differences have been noticed. Lrit3-/- mice exhibit a slight decrease in thickness of the inner nuclear layer [34], while such an observation was absent in retinas of Gpr179-/- mice [4].
[136-137] “suggesting that”: Here, for once, you can be more assertive – or simply delete the second half of the sentence, which is just another way of saying what the first part says.
We thank the reviewer for this correction. “Suggesting that” was replaced by “therefore”, line 176.
[Figure 3] The graphs are generally clear and informative. In this figure, though, I suggest making it even easier to read by placing these keys to the right of their respective 3-week points.
We thank the reviewer for this idea to make the graph clearer. We agree with this point. Changes have been made in figure 3.
[173-174] Here at last is the necessary statement as to how the ON-pathway affect DA metabolism. I still feel that it should be added to the Introduction, although maybe keeping it here as well.
We agree with the reviewer about the necessity of highlighting how the ON pathway acts towards DA turnover and metabolism. As suggested, we write in the current version of the introduction, lines 70-78: “Indeed, in the retina, DA is synthesized by dopaminergic amacrine cells (DACs), its secretion is regulated by light conditions [41-44] and requires a functional ON-pathway [29, 45]. The retinal level of the DA degradation metabolite, 3,4 dihydroxyphenylacetic acid (DOPAC) is a commonly used marker of DA release and the DOPAC/DA ratio is considered a marker of DA turnover. It was shown that DA and DOPAC levels are reduced in a variety of myopia experimental models including FDM and LIM [17, 43, 44, 46-48] and experimentally increased retinal DA levels blocks FDM and LIM in amphibians [42], mammals [43, 44] and avian [48, 49], and vice versa [38, 40]”
A paragraph in the discussion was also modified as follow: lines 207-214: “Functional GPR179 is necessary for a normal ON signaling, and DACs receive indirect excitatory input from ON-BCs. Furthermore, stimulation of the ON pathway induces the release of DA via the activation of DACs [42, 52, 98]. Thus, it has been hypothesized that defects in ON-BCs signaling could impact DA release [29, 45, 99]. Our results are in line with previous studies addressing this hypothesis [29, 45].”
[175-176] I think that the difference in effects on retinal DA metabolism, in GRM6 vs GPR179 models, is an important one. (a) The discussion of differences in lighting and treatment time as possible causes for differences in outcomes is relevant; but if you really think that these might be the cause, why didn’t you (or why don’t you now) add experiments that duplicate some of these conditions in your model? (b) You might also consider the possibility that both of these genes are expressed in somewhat different subpopulations of mGluR-expressing cells. At least mention that this is a possibility, but note that “GPR179 is not expressed elsewhere in the retina” (Peachey et al.. 2012, Ref.#63); is that also true of mGluR6, or not?. [183-192] This is a good argument for the likelihood that an ON-pathway deficit is responsible for susceptibility to myopia induction. It doesn’t rule out other possibilities, however, nor does it explain the discrepancy between GRM6 and GPR179 mice w.r.t. retinal DA/DOPAC.
We are grateful to the reviewer for this interesting point. However, we think our and other studies on cCSNB mouse models and myopia reflect a similar outcome, namely altered dopamine metabolism and higher susceptibility to induced myopia and this is the major point we would like to make in our article. (a) We consider that the small differences obtained using different cCSNB can be due to technical issues, or indeed highlight the differences of the different roles of each gene, deleted in each cCSNB model. In fact, we had difficulties to explain the differences in DA metabolism between the two models. Performing experiments to check the impact of lightning environment upon retinal DA and DOPAC levels would be of interest. But our animal facility only have rooms in complete darkness or at a controlled illuminance of 50 lux, testing different lightning conditions would be difficult. Another point that we can’t control nor modify for now is about the housing of animals. Our cages are based on a standard model which might not be the best one to perform such comparisons. Nevertheless, this is a very interesting idea and we will address this possibility in future experiments. (b) We have assumed that Grm6 and Gpr179 are expressed in the same neuronal subpopulations in the retina. But to be more cautious and in accordance with the suggestions of the reviewer we write in the novel version of the manuscript: line 222 Another explanation of the observed differences might be due to a different expression or protein localization of GRM6 and GPR179. However, to our knowledge, both transcripts are specifically expressed in ON-BCs [40] and localize specifically in their dendritic tips [41]. We added a small paragraph to the discussion section addressing this point: lines 232-238.: “A more interesting explanation might be that although both GRM6 and GPR179 are necessary for ON-BC signaling and are thought to be expressed in all ON-BCs and similarly localized to their dendritic tips, it is not known if the physiological consequences of their absence is the same. Indeed, the light responses of retinal ganglion cells (downstream of ON BCs) are different in mice with different Grm6 mutant alleles [83], thus it is possible that DACs are differentially affected in Grm6-/- and Gpr179-/- mice “
[196-198] I think that it would be helpful to add further explanation here – e.g., what you mean by "hypothesis of an impact of ON-pathway deficit", and more specifically, how the observed DA and DOPAC data support it. [199-220] Again, the extensive speculation (some of which repeats earlier discussion, and it would be good to consolidate these sections) begs the question: Why didn’t you attempt to replicate the conditions of previously published experiments on GRM6-KO mice, rather than striking out on your own and using a unique protocol w.r.t. lighting, age, and treatment duration?
We thank the reviewer for pinpointing the lack of details in this point and precise in the current version of the manuscript we, line 243: “Overall, our results are in line with the hypothesis of a defective ON-pathway causing a reduction of retinal dopamine.”
Please find below the reasoning of our chosen protocols: For the DA/DOPAC: for lightning conditions, as mentioned above, our animal facility only have a controlled room at 50 lux (used for the light adapted condition in DA/DOPAC experiments) and a dark room (used for dark adaptation in DA/DOPAC and ERG experiments). About the timing of light adaptation, we used 4 hours as mentioned in the Nyx paper (MT Pardue & al.; 2008). The Grm6 paper does not give any information about the timing of light adaptation but as they both come from the same lab, we assumed it would be equal times. About the age of tested mice in DA/DOPAC experiments, we used the same age as in the Nyx paper. Furthermore, 12 weeks old mice (as done in the Nyx paper) are easier to generate than 16 weeks old (as done in the Grm6 paper).
For the LIM experiments: When we started to develop the myopia induction protocol, we did not know if we will have a decrease in DA/DOPAC levels (induced myopia experiments and DA/DOPAC measurements were performed in parallel). According to “An updated view on the role of dopamine in myopia” from M Feldkaemper and F Schaeffel, 2013, LIM protocol is thought to be less dependent of dopaminergic signaling (even if it is clearly not independent either). Furthermore, LIM is often considered as a more relevant protocol to study emmetropisation process (even if it is also more complicated to interpret) as it takes also in account the sign and extent of imposed defocus (“Form deprivation and lens-induced myopia: are they different? I. Morgan et al., 2013”). As mentioned in the material and methods section, we adapted our protocol from X. Jiang et al.; 2018: A highly efficient murine model of experimental myopia. In this study, the authors propose a fast and strong protocol to induce myopia with facilitated way to remove and clean the lenses. Inducing myopia in 3 weeks seemed technically more affordable than in 6 or 8 weeks (as observed in the Nyx and Grm6 papers). Finally, in mice, many data are available using the FDM protocol, much less with the LIM protocol and none about LIM in models of ON-pathway defect.
[234] “consequent to” is, I think, the wrong term; try “"due to", or "consequent to".
We thank the reviewer for this correction, “consequent (or subsequent) to” was replaced by “due to”, line 333
*A brief note here: In general the text is clear and readable, but little errors in English grammar, usage, and punctuation are scattered throughout it; for me, this was negative and distracting. It would be kind to readers (and the journal’s editorial staff), as well as a “good show” for the authors, to have an English-speaking specialist edit the language carefully before submission.
We are grateful to the reviewer for highlighting the weaknesses of English grammar in our manuscript. The final version of the manuscript has been corrected by English native speaker.
[236] "confirm" seems to me to be too strong; I would suggest replacing it with "support" or "are consistent with". 4 *I have not critiqued the Methods in detail, because I trust the authors to rephrase correctly what they have published before. The writing should receive editorial attention, though. Afew further comments below.
We thank the reviewer for the precision. We considered that we could use the term “confirm” as it is already the third paper (and using a third model and another protocol of myopia induction) showing an impact of ON-pathway defect upon dopaminergic signaling and myopia susceptibility. But in order to soften this, we replaced “confirm” by “strengthen” line 308.
[285] “A skull scalp was performed”: A more specific explanation would be preferred.
We gave more detailed information by rephrasing: lines 389-391: “The scalp was cut to expose 1.2 cm of the skull in the rostrocaudal axis. Two intracranial screws were implanted on both left and right sides of the skull at y = -2 mm from the bregma”
[510-512] Ref.#69, Pardue et al., duplicates Ref.#29.
We are grateful to the reviewer for this correction. Changes were made in the references to remove the duplicated ref.
*I think that it would be particularly interesting to record from ON-BCs in mouse retinal slices, to determine whether the engineered mutations cause these cells to be not only unresponsive to changes in synaptic signalling from the photoreceptors, but also cause them to be constitutively “active” (depolarized) or “inactive” (hyperpolarized). It’s worthy of note that the mGluR6 antagonist L-AP4 (APB), inhibits axial elongation and causes a hyperopic shift in otherwise untreated chicken eyes, but has no effect on the development of form-deprivation myopia (Fujikado et al., 1996 Curr Eye Res, 15(1):79-86, doi: 10.3109/02713689609017614; and Crewther et al., 1996 J Ocul Pharmacol Ther, 12(2):193-208. doi: 10.1089/jop.1996.12.193). APB is an mGluR6 agonist that makes the ON-BCs unresponsive to L-Glu (and thus to changes in photoreceptor activity) by binding almost irreversibly to mGluR6 and holding it in the activated (therefore hyperpolarized) state. One might conclude, therefore, that the absence of L-Glu release from ON-BCs causes a decrease in axial elongation and shift towards hyperopia, and therefore that depolarization of ON-BCs in light would cause increased axial elongation and a shift towards myopia. In the absence of mGluR6 or GPR179, the ON-BCs should be constitutively depolarized, and the effects on refractive development should be opposite the effects of APB; that is, deletion of these genes should cause myopia. You might consider adding some discussion of this to the paper, or at least giving it serious thought – and pondering why APB inhibits axial elongation and myopia in normally-seeing eyes, but not in form-deprived ones. ...
We thank the reviewer for addressing those interesting points of discussion. We agree that it would be of interest to perform patch clamp recordings on cCSNB models ON-BCs and compare their spontaneous active state to the wild-type littermates. The fact that injection of L-AP4 induces a decrease in axial elongation and a consequent hyperopic shift in non treated chicken eyes but not in form-deprived ones are remarkable data. To address this question, the experiment you proposed would be relevant. One can think that another actor of retinal signaling implicated in dopaminergic signaling, circadian rhythm, refractive development and thus, myopia, without being directly implicated in ON-pathway might counteract the effect of L-AP4 upon FDM. Thus, I would add that it might be interesting to challenge those putative factors.
For instance, upstream of the synaptic signaling between photoreceptors and ON-BCs, we can note the release of melatonin, acting in an opposite way to dopamine with both forming mutual inhibitory feedbacks to each other. Melatonin levels seems to be unchanged during form deprivation in chicken, but injecting melatonin has an impact upon FDM and ocular growth (See Hoffmann M, Schaeffel F. Melatonin and deprivation myopia in chickens. Neurochem Int. 1996 Jan;28(1):95-107. doi: 10.1016/0197-0186(95)00050-i. PMID: 8746769. and Rada JA, Wiechmann AF. Melatonin receptors in chick ocular tissues: implications for a role of melatonin in ocular growth regulation. Invest Ophthalmol Vis Sci. 2006 Jan;47(1):25-33. doi: 10.1167/iovs.05-0195. PMID: 16384940.). This is consistent with results found on C57BL/6 mice (which are melatonin-deficient): FDM in those mice does not induce any change in DA levels, but experimentally-induced changes in DA levels do have an impact upon FDM. In contrast, inducing FDM in melatonin-proficient mice (CBA/CaJ) leads to altered retinal DA levels (Qian KW, Li YY, Wu XH, Gong X, Liu AL, Chen WH, Yang Z, Cui LJ, Liu YF, Ma YY, Yu CX, Huang F, Wang Q, Zhou X, Qu J, Zhong YM, Yang XL, Weng SJ. Altered Retinal Dopamine Levels in a Melatonin-proficient Mouse Model of Form-deprivation Myopia. Neurosci Bull. 2022 Sep;38(9):992-1006. doi: 10.1007/s12264-022-00842-9. Epub 2022 Mar 27. PMID: 35349094; PMCID: PMC9468212.). Besides, this could partly explain the residual retinal levels of DA observed in cCSNB mouse models (they are under a melatonin-deficient genetic background).
Thus, a good, though ambitious way to address the potential role of melatonin on the differential effect of L-AP4 upon form-deprived and untreated eyes would be to generate a cCSNB mouse model in a CBA/CaJ genetic background (or any other melatonin-proficient background) and compare it to a cCSNB mouse model under C57BL/6J genetic background (or any other melatonin-deficient background).
If melatonin is the factor (or one of the factors) counteracting the effect of L-AP4, then its absence in C57BL/6J wild-type mice injected with L-AP4 should not cause a hyperopic shift. FDM in these mice should be similar to those from C57BL/6J KO (that is, stronger myopic shift than in goggled C57BL/6J wild-type uninjected) as L-AP4 hyperpolarizes ON-BCs and thus inactivates the ON-pathway. In contrast, in CBA/CaJ KO mice, the presence of melatonin and the inactivated ON-pathway might lead to a higher myopic shift –maybe even higher than in C57BL/6J KO due to the melatonin-mediated inhibition of dopamine release? Obviously, measuring retinal levels of DA and DOPAC in all those different conditions might be of interest too.
Another (perhaps easier) way to address this question would be to compare the impact of L-AP4 injections upon normal refractive development, FDM condition –thought to be a “constant condition” which might alter the timing cue of lighting, a key regulator of the eye growth (see Morgan IG, Ashby RS, Nickla DL. Form deprivation and lens-induced myopia: are they different? Ophthalmic Physiol Opt. 2013 May;33(3):355-61. doi: 10.1111/opo.12059. PMID: 23662966; PMCID: PMC3745013.) – and LIM in melatonin-deficient and melatonin-proficient mice. As LIM is supposed to not have an impact upon the timing cue (or a reduced one), WT CBA/CaJ mice with LIM and injected with L-AP4 might show a slower or lower myopic shift compared to wild-type CBA/CaJ mice with FDM injected with L-AP4.
Ideas for further experiments, perhaps? ?
Indeed, as our work validate the potential use of cCSNB models to study myopia, we think they can be used to identify new genes related to myopia in cCSNB. For instance, by performing RNAseq studies upon cCSNB models and patients, we may find genes implicated in syndromic and/or common myopia
Furthermore, the two last commentaries of the reviewer enabled us to rewrite the end of our discussion section:
We need to elucidate if the myopic shift induced by our LIM protocol directly leads to a decrease in retinal levels of DA and/or DOPAC in wild-type and in Gpr179-/- mice, Interestingly, recent studies showed that injecting DA agonists suppresses FDM [48, 116, 117]. In addition, induction of myopia by FDM seemed to decrease retinal levels of DA in guinea pigs [118], chickens [119] and primates [91] but not in wild-type mice [120]. This may be at least partly explained by the fact that most mice used in myopia studies are melatonin-deficient mice (i.e. C57BL/6J mice). Melatonin is released by photoreceptors, is implicated in circadian rhythm, and seems to inhibit dopamine release. Dopamine and melatonin act in opposite ways, interacting with each other to form an inhibitory feedback loop [121]. In support of the notion that melatonin is required for FDM-related retinal DA changes, a recent study showed that melatonin-proficient mice (i.e. CBA/CaJ) display reduced retinal levels of DA during FDM [122]. Similarly, this may explain a part of the residual dopamine release in the retinas of cCSNB mouse models observed in our study and in others [29, 45] Further experiments are required to determine whether the causal link between DA metabolism and sensitivity to myopia induction is unidirectional or bidirectional in mice. Mechanisms implicated in LIM-mediated myopic shift seem to be less dependent on DA metabolism than those mediated by FDM [43, 47], but are clearly not independent either. This can be explained by the loss of timing cue of lighting occurring during FDM but not during LIM [47].
It is noteworthy that injecting L-AP4 in chicken causes a hyperopic shift in ungoggled eyes but does not have an impact on form-deprived ones [123, 124]. L-AP4 is an agonist of group III metabotropic glutamate receptors. In the retina, it stimulates mGluR6 and whether it causes the ON-BCs to be constitutively hyperpolarized is still unknown. To our knowledge, there is still little evidence on the resting state of ON-BCs in cCSNB models. Whole cell patch-clamp recordings of RBCs from nob mice retinal slices revealed small holding currents, indicating that channels were mostly closed or absent [125]. Further studies focusing on the differential effect of L-AP4 upon ungoggled eyes and form-deprived ones might be of interest to decipher the mechanisms leading to myopia in cCSNBs.
Altogether, our results indicate that the loss of GPR179 induces disturbances in the dopaminergic system. The ON-pathway defect due to the lack of GPR179 does not have any measurable impact upon refractive development but increases the sensitivity to LIM. Overall, our data strengthen the hypothesis that ON-pathway deficiency can be associated with myopia onset through alteration of dopaminergic signaling. This work validates the potential use of cCSNB models to study myopia. Thus, we think they can be used to identify new genes related to myopia in cCSNB. For instance, by performing RNAseq studies upon cCSNB models and patients, we may find genes implicated in syndromic and/or common myopia.
Reviewer 3 Report
Myopia is a public health problem globally. Dr. Zeitz and her colleagues reported the current research on a novel myopia mouse model. The mouse with lack of Gpr179 and impaired ON-pathway lead to myopia with the alteration of the dopaminergic system. This work no doubly focused on an important scientific question.
Recently increasing research found that dopamine and ON pathway involved myopia development. The manuscript will help interested researchers in the myopia area.
I had the following questions, suggestions, and comments in line with the work.
1: The comparison of retinal levels of DA; DOPAC and ratio, could you explain why use 4 hours after light adaptation not 2 hours or even 1 hours? or the 6 hours?
2: The mice of P21, P35 and P42 were used. It will be better if the retinal levels of DA; DOPAC and ratio during that time be listed if authors had.
In summary, I think the work is well written logically and will benefit myopia research.
Author Response
Dear Dr. Spasojevic, dear reviewers,
We are grateful for the invitation to submit a revised version of our manuscript “Mice lacking Gpr179 with complete congenital stationary night blindness are a good model for myopia”, ID Ref.: ijms-1980861. We are also thankful for the fruitful comments, which helped us improve the quality of the manuscript. Enclosed we return the revised version. We have responded to all criticisms and suggestions and we list below how and where we have implemented the changes to the new version of the manuscript. In addition, all other changes are visualized by “track-changes”. Thank you for your patience!
Yours sincerely,
Baptiste Wilmet and Christina Zeitz
Reviewer 3:
Comments and Suggestions for Authors
Myopia is a public health problem globally. Dr. Zeitz and her colleagues reported the current research on a novel myopia mouse model. The mouse with lack of Gpr179 and impaired ON-pathway lead to myopia with the alteration of the dopaminergic system. This work no doubly focused on an important scientific question.
Recently increasing research found that dopamine and ON pathway involved myopia development. The manuscript will help interested researchers in the myopia area.
We are grateful to the reviewer for showing interest about our work and for the relevant question and comments.
I had the following questions, suggestions, and comments in line with the work.
1: The comparison of retinal levels of DA; DOPAC and ratio, could you explain why use 4 hours after light adaptation not 2 hours or even 1 hours? or the 6 hours?
We thank the reviewer for addressing this interesting question. We used 4 hours of light adaptation to ensure that 1) Light-induced mechanisms of DA release are completed and 2) to replicate the parameters tested in MT Pardue et al., 2008 on Nyx model and in R. Chakraborty et al., 2015 on Grm6 model.
2: The mice of P21, P35 and P42 were used. It will be better if the retinal levels of DA; DOPAC and ratio during that time be listed if authors had.
We agree with the reviewer that obtaining those data would be of interest. We did not perform these experiments for several reasons: First, the HPLC measurements of DA/DOPAC are not performed in our lab at the Institut de la Vision but in the laboratory of Jacques Callebert, at the Larboisiere hospital. Measuring DA and DOPAC at P21, P35 and P42 would have required much more mice as those are end point experiments requiring the euthanasia of animals. A paragraph was added in the discussion to address this question. Furthermore, recent studies showed that injecting DA agonists suppresses FDM [48, 116, 117]. In addition, induction of myopia by FDM seemed to decrease retinal levels of DA in guinea pigs [118], chickens [119] and primates [91] but not in wild-type mice [120]. This may be at least partly explained by the fact that most mice used in myopia studies are melatonin-deficient mice (i.e. C57BL/6J mice). Melatonin is released by photoreceptors, is implicated in circadian rhythm, and seems to inhibit dopamine release. Dopamine and melatonin act in opposite ways, interacting with each other to form an inhibitory feedback loop [121]. In support of the notion that melatonin is required for FDM-related retinal DA changes, a recent study showed that melatonin-proficient mice (i.e. CBA/CaJ) display reduced retinal levels of DA during FDM [122]. Similarly, this may explain a part of the residual dopamine release in the retinas of cCSNB mouse models observed in our study and in others [29, 45] Further experiments are required to determine whether the causal link between DA metabolism and sensitivity to myopia induction is unidirectional or bidirectional in mice. Mechanisms implicated in LIM-mediated myopic shift seem to be less dependent on DA metabolism than those mediated by FDM [43, 47], but are clearly not independent either. This can be explained by the loss of timing cue of lighting occurring during FDM but not during LIM [47]
We are discussing the inclusion of DA/DOPAC measurements during LIM protocols in our future studies.
In summary, I think the work is well written logically and will benefit myopia research.
We are grateful to the reviewer for supporting our work.
Round 2
Reviewer 2 Report
I commend you for your excellent and appropriate responses to my review, which have resulted in a very nice report.
While the journal's editorial staff will proof-read the writing further, I do suggest that you ask your English-language colleague to have one more go at it. Here are some examples of the kinds of things that could be improved upon [line numbers in brackets]:
[54]: "genetic" -> "genetics"
[65]: add "of which" at the end, after "both"
[72]: insert "it" before "requuires"
[78]: levels "block"
[99]: I suggested "signs", instead of "symptoms", thinking that symptoms are what the patient feels, whereas signs are what the examiner observes. But I might be wrong, since I haven't seen a patient or worked in a clinical setting since finishing med school.
[140]: insert "some subtypes of", before 'most major classes'
[181]: "kimetic" -> "kinetics"
[193]: "ns" doesn't appear in the graph, so last 3 words can be deleted
I like the way Fig. 2 looks now.
[229]: delete 2nd word, "a"
[237]: insert: "in Gpr179-KO mice," between "reduced ... while" ...
[245-251]: Good! I like the way this argument was developed.
[310]: "if" -> "whether"
[313]: insert: "in several other animal models" after 'suppresses FDM'
[313]: instead of "myopia by FDM" use either: "FDM", or "myopia by FD" (FDM is the result of FD, not what induces myopia.
[336-338]: Good addition!
Author Response
Dear all,
We are grateful for the invitation to submit a revised version of our manuscript “Mice lacking Gpr179 with complete congenital stationary night blindness are a good model for myopia”, ID Ref.: ijms-1980861. We are thankful for the additional fruitful comments, to improve the quality of the revised version of our manuscript. Enclosed we return the revised version. We have responded to all criticisms and suggestions and we list below how and where we have implemented the changes to the new version of the manuscript. In addition, all other changes are visualized by “track-changes”. Thank you for your patience!
Yours sincerely,
Baptiste Wilmet and Christina Zeitz
I commend you for your excellent and appropriate responses to my review, which have resulted in a very nice report.
We are glad that the reviewer appreciated our responses.
While the journal's editorial staff will proof-read the writing further, I do suggest that you ask your English-language colleague to have one more go at it. Here are some examples of the kinds of things that could be improved upon [line numbers in brackets]:
We are thankful for the time this reviewer spent to improve our English language, double-checked it again by an English-language colleague and made additional changes as suggested below:
[54]: "genetic" -> "genetics"
[65]: add "of which" at the end, after "both"
[72]: insert "it" before "requuires"
[78]: levels "block"
[99]: I suggested "signs", instead of "symptoms", thinking that symptoms are what the patient feels, whereas signs are what the examiner observes. But I might be wrong, since I haven't seen a patient or worked in a clinical setting since finishing med school.
[140]: insert "some subtypes of", before 'most major classes'
[181]: "kimetic" -> "kinetics"
[193]: "ns" doesn't appear in the graph, so last 3 words can be deleted
I like the way Fig. 2 looks now.
[229]: delete 2nd word, "a"
Unfortunately, the numbering of the lines were different in our document, so we are not sure which the reviewer is referring to. We did however put in the plural “experiments revealed drastic decreases of both…”
[237]: insert: "in Gpr179-KO mice," between "reduced ... while" ...
[245-251]: Good! I like the way this argument was developed.
[310]: "if" -> "whether"
[313]: insert: "in several other animal models" after 'suppresses FDM'
[313]: instead of "myopia by FDM" use either: "FDM", or "myopia by FD" (FDM is the result of FD, not what induces myopia.
[336-338]: Good addition!
In addition, Figure 1 was replaced to make to homogenize the writing style.